DisCoVering potential candidates of RNAi-based therapy for COVID-19 using computational methods

Rohani Narjes 1 2
Ahmadi Moughari Fatemeh 1
http://orcid.org/0000-0002-8913-3904 Eslahchi Changiz 1 2 ch-eslahchi@sbu.ac.ir
1 Department of Computer and Data Sciences, Faculty of Mathematical Sciences, Shahid Beheshti University , Tehran , Iran
2 School of Biological Sciences, Institute for Research in Fundamental Sciences (IPM) , Tehran , Iran
Fischer Daniel
Electronic publication date: 2021 Feb 26
Publication date: 2021
Volume: 9
Electronic Location ID: e10505
Received 2020 Aug 27; Accepted 2020 Nov 15
Copyright: © 2021 Rohani et al.
Copyright year: 2021
Copyright holder: Rohani et al.
License: This is an open access article distributed under the terms of the Creative Commons Attribution License, which permits unrestricted use, distribution, reproduction and adaptation in any medium and for any purpose provided that it is properly attributed. For attribution, the original author(s), title, publication source (PeerJ) and either DOI or URL of the article must be cited.
License URL: https://creativecommons.org/licenses/by/4.0/

Keywords: COVID-19, SARS-CoV-2, RNA intervention-based therapy, miRNA, miRNA-mRNA interaction, siRNA design, Virology

Funding: The authors received no funding for this work.

==============================
The ongoing pandemic of a novel coronavirus (SARS-CoV-2) leads to international concern; thus, emergency interventions need to be taken. Due to the time-consuming experimental methods for proposing useful treatments, computational approaches facilitate investigating thousands of alternatives simultaneously and narrow down the cases for experimental validation. Herein, we conducted four independent analyses for RNA interference (RNAi)-based therapy with computational and bioinformatic methods. The aim is to target the evolutionarily conserved regions in the SARS-CoV-2 genome in order to down-regulate or silence its RNA. miRNAs are denoted to play an important role in the resistance of some species to viral infections. A comprehensive analysis of the miRNAs available in the body of humans, as well as the miRNAs in bats and many other species, were done to find efficient candidates with low side effects in the human body. Moreover, the evolutionarily conserved regions in the SARS-CoV-2 genome were considered for designing novel significant siRNA that are target-specific. A small set of miRNAs and five siRNAs were suggested as the possible efficient candidates with a high affinity to the SARS-CoV-2 genome and low side effects. The suggested candidates are promising therapeutics for the experimental evaluations and may speed up the procedure of treatment design. Materials and implementations are available at: https://github.com/nrohani/SARS-CoV-2.

Introduction

Recently a new coronavirus (CoV) named severe acute respiratory syndrome coronavirus 2 (SARS-CoV-2) has emerged from China and globally outbreak (Saini et al., 2020) with over 48 million cases and over 1 million deaths (as the last update in November 5, 2020). World Health Organization (WHO) has warned that this outbreak is pandemic and demands for emergency researches and studies to find efficient treatment strategies against it.

COVID-19 infection causes multiple frequent symptoms such as fever, dry cough, difficulty breathing or shortness of breath, dyspnea, fatigue, headache, diarrhea, and lymphopenia (Rothan & Byrareddy, 2020; Wu et al., 2020a). Individuals with severe infection are at the risk of the severe acute respiratory syndrome, kidney failure, pneumonia, and, unfortunately, death (Wu et al., 2020a). SARS-CoV-2 is in the taxonomy of Betacoronavirus. This family includes seven species, namely HCoV-HKU1, HCoV-229E, HCoV-NL63, HCoV-OC34, SARS-CoV, MERS-CoV, and SARS-CoV-2 (Khan et al., 2020). Humans and vertebrates are susceptible to infections of coronavirus. Nevertheless, no vaccine has been approved for SARS-CoV-2 or other human coronavirus (Wu et al., 2020a). SARS-CoV-2 genome is a single-stranded positive-sense of 29,903 nucleotides (nt) length (Ahmed, Quadeer & McKay, 2020), which has 89.1% and 60% of sequence in common with SARS and MERS, respectively (Khan et al., 2020). The SARS-CoV-2 sequence contains ten open reading frames (ORFs) (Khan et al., 2020) and encodes for structural (spike, envelope, membrane, and nucleocapsid) proteins as well as non-structural proteins (nsp) (Ahmed, Quadeer & McKay, 2020).

Because of the recent outbreak of COVID-19, our knowledge about its pathogenesis, molecular mechanisms, prevention, and treatment strategies is deficient (Ahmed, Quadeer & McKay, 2020). Besides the hectic efforts for proposing vaccines against this disease, numerous approaches for its treatment are used. These approaches can be categorized into four groups: viral replication and translation inhibition, impeding viral-host receptor binding, improving the innate immunity of the host, and blocking specific enzymes or receptors in host (Han & Král, 2020). Providing new insights about the molecular mechanism of SARS-CoV-2 and its conserved regions can give a clue to propose efficient treatment.

We aim to prevent virus activities through inhibiting viral replication and translation, or by prohibiting the viral-host binding. To this aim, we target the most functional regions that are responsible for viral replication and translation, or viral-host binding. The evolutionarily conserved regions in SARS-CoV-2 have more potential to play critical roles in viral replication and translation because these regions were selected and conserved during evolution. Therefore, these regions are promising candidates for targeting by antiviral and oligonucleotide therapies (Rangan, Zheludev & Das, 2020). Moreover, the viruses may be disinclined to reveal resistance against the treatments that target the conserved regions since these regions are probably vital for the functionality of the virus (Rangan, Zheludev & Das, 2020). Among the conserved regions, unstructured parts are more compelling since they had more inclination to bind to oligonucleotide therapies via hybridization. Thus, the unstructured conserved regions are more favorable targets due to its conservation, disinclination to develop resistance, high affinity to bind by hybridization and more accessibility to therapeutic interventions (Rangan, Zheludev & Das, 2020).

RNA interference (RNAi) is an alternative therapeutic strategy when current treatment technologies fail to obtain a promising result (Bobbin & Rossi, 2016). RNAi-based therapy is especially proficient for the treatment of viral infections, which escape other strategies due to their mutations (Bobbin & Rossi, 2016). Nowadays, big pharmacology companions consider RNAi-based therapy in their clinical trial trends (Chakraborty et al., 2017). RNAi delivers small RNA duplexes such as microRNA (miRNA) or short interfering RNA (siRNA) in the body to start inhibiting specific genes (Hannon, 2002; Setten, Rossi & Han, 2019). siRNAs are completely complementary to their targets, while miRNAs are partially complementary; therefore, miRNAs repress the translation of their targets, whereas siRNAs lead to Argonaute 2-mediated degradation (Bobbin & Rossi, 2016).

miRNAs are small non-coding RNAs with usually 22 nt lengths, which have vital functions in post-transcriptional regulation of target genes (Witkos, Koscianska & Krzyzosiak, 2011). They play a role in cell cycle progression, cell differentiation, cell proliferation, apoptosis (Tabas-Madrid et al., 2014), and numerous cellular processes that are vital for triggering or adaptive immunity (Drury, O’Connor & Pollard, 2017). miRNAs are primarily used to decline the mRNA level of their targets. This mechanism is done through imperfect binding to the target sites and causing either inhibition in translation or RNA cleavage (Witkos, Koscianska & Krzyzosiak, 2011; Muniategui et al., 2012). miRNAs made a paradigm shift in our insights about gene regulation and therapeutic strategies. miRNAs are one of the efficient approaches in designing therapies for silencing or downregulating pathogenesis mRNAs (Witkos, Koscianska & Krzyzosiak, 2011). Utilizing miRNAs in clinical interventions for combating infectious viruses has initiated 24 years after discovering the first miRNA (Drury, O’Connor & Pollard, 2017). Nowadays, numerous novel vaccines are curated using miRNAs that targeted viral genomes (Drury, O’Connor & Pollard, 2017; Shen et al., 2015; Heiss, Maximova & Pletnev, 2011; Tan et al., 2016; Brostoff et al., 2016; Perez et al., 2009). The virus attenuation is done by downregulating the mRNA levels of the virus, which is a highly efficient approach and reduces the harm (Drury, O’Connor & Pollard, 2017).

siRNA has high specificity and is an efficient strategy for suppressing specific genes (Tai & Gao, 2017). Since siRNAs can target and silence essential genes in the virus survival, using siRNAs has been approved as a promising therapeutic against viral infections (Li et al., 2005). Currently, several siRNAs have been proposed for inhibiting viral replication for poliovirus and Rous sarcoma (Gitlin, Karelsky & Andino, 2002; Hu et al., 2002), human immunodeficiency virus (HIV) (Coburn & Cullen, 2002; Martnez, Clotet & Esté, 2002; Jacque, Triques & Stevenson, 2002; Park et al., 2002), hepatitis C (HCV) (Kapadia, Brideau-Andersen & Chisari, 2003; Yokota et al., 2003) and hepatitis B (HBV) (Shlomai & Shaul, 2003; McCaffrey et al., 2003) viruses. Previous studies on cultured cells with SARS-CoV have revealed promising results of using siRNA in therapies for inhibiting viral replication (He et al., 2003; Zhang et al., 2004; Lu et al., 2004; Li et al., 2005).

Nevertheless, suggesting candidate miRNAs and siRNAs that potentially have high affinities to interact with the SARS-CoV-2 genome is a promising idea that requires significant attention. One of the challenges in using RNAi-based treatments is the competition between the body mRNAs and SARS-CoV-2 RNA for binding to the miRNA/siRNA (Demirci, Yousef & Allmer, 2019). If miRNA/siRNA tends to bind body mRNAs instead of SARS-CoV-2 RNA, this issue may lead to a decline in treatment efficiency or to develop side effects. Therefore, it is essential to investigate the interactions between miRNAs and human mRNAs and analyze the affinity of binding the suggested miRNAs with other mRNAs instead of SARS-CoV-2 RNA.

Several databases have gathered the experimentally-validated miRNA-mRNA interactions (Tabas-Madrid et al., 2014; Helwak et al., 2013; Licatalosi et al., 2008; Dweep et al., 2011). Moreover, computational approaches have sped up miRNA-mRNA interaction prediction by facilitating the rapid analysis of thousands of data (Al-Khafaji, AL-DuhaidahawiL & Taskin Tok, 2020). The important elements in the efficient interaction between miRNA and its targets are free binding energy, site accessibility, seed match, and evolutionary conservation (Peterson et al., 2014). All of these constraints are incorporated in this study.

Besides the miRNAs available in the human body, the miRNAs available in other species can have a promising mechanism against SARS-CoV-2. Among all species, bats (Chiroptera) are the most favorable, since bats are the most probable origin of SARS-CoV-2 and some other CoVs (Lai et al., 2020; Zhou et al., 2020). Moreover, bats host numerous pathogenic viruses such as paramyxo, rhabdo-, filoviruses (such as Ebola and Marburg viruses), lyssaviruses, coronaviruses, and henipaviruses (e.g., Hendra and Nipah viruses) (Hoffmann et al., 2013; Slater, Eckerle & Chang, 2018). Meanwhile, bats have shown high resistance to viral infections and reveal no symptom or mild symptoms to this deadliest diseases (Slater, Eckerle & Chang, 2018). Therefore, the miRNAs in the bat body may give us a clue to find efficient options for the treatment of COVID-19. In addition to bats, many other species may contain miRNAs that may be competent candidates.

This study aims to target the unstructured conserved regions (UCRs) of SARS-CoV-2 using the available miRNAs or novel siRNAs. In this study, some candidate miRNAs from human, bat, and other species are suggested that potentially inhibit SARS-CoV-2 replication and translation while having less affinity to bind other human body mRNAs. Using these miRNAs aids in maximizing treatment efficiency and minimizing side effects. The affinity binding of miRNAs to the UCRs of SARS-CoV-2 is computed to ensure a high potential treatment. To further investigate the effects of using suggested miRNAs, the biological functions of their targets in UCRs of SARS-CoV-2 are analyzed. Moreover, the potential side-effects are investigated by examining the functionalities of mRNAs that may computationally interact with the suggested miRNAs. In addition to recommending efficient miRNAs, five efficient and target-specific siRNAs are proposed for silencing SARS-CoV-2 genes. The suggested siRNAs satisfy in the required sequence constraints (as described in Ui-Tei et al. (2004)), have low off-target effects due to their thermodynamic stability, and are not likely to target other mRNAs in the human body. The suggested miRNAs and siRNAs are promising candidates for experimental validations.

Materials and Method

Datasets

The complete genome sequence of SARS-CoV-2 was obtained with accession No. NC045512.2 from the GenBank database (Benson et al., 2012; Wu et al., 2020b). The complete genome of betacoronavirus sequences from the NCBI database and sequences compiled by Ceraolo & Giorgi (2020) were considered for finding evolutionarily conserved regions.

In this study, we conducted three independent analyses for suggesting candidate miRNAs and one analysis for designing efficient siRNAs. The miRNAs and their interaction information in theses analysis are obtained from the following sources:In the first analysis (Analysis #1), the list of miRNAs and their interactions were collected from CLASH (Helwak et al., 2013). This database comprises 7,390 experimentally validated interactions between 270 miRNAs and 7,390 mRNAs (Plotnikova, Baranova & Skoblov, 2019). This database contains the free binding energy of interactions as well as the sequences of miRNAs and mRNAs involving in the interactions.

In the second analysis (Analysis #2), the sequences of 539 bat-specific miRNAs were obtained from the previously published article by Huang, Jebb & Teeling (2016). Since the interaction between bat-specific miRNAs and human mRNAs are not provided experimentally, we extracted the human mRNA targets for bat-specific miRNA using mirDB (Chen & Wang, 2020) tool.

In the third analysis (Analysis #3), the known miRNAs of 286 species were downloaded from the miRNA registry, mirBase (Kozomara, Birgaoanu & Griffiths-Jones, 2019). mirBase is the most comprehensive database for miRNAs (Drury, O’Connor & Pollard, 2017) and comprises 6,133 miRNAs. However, it does not contain the miRNA-mRNA interactions. We used mirDB (Chen & Wang, 2020) tool for obtaining the human mRNA targets for candidate miRNAs from mirBase.

The fourth analysis (Analysis #4) is conducted for proposing new potential siRNAs, for which siDirect 2.0 (Naito et al., 2009) tool is used.

Methodology

All analyses in this study are conducted with the aim of either recommending potential miRNAs or proposing efficient siRNAs. These two procedures are described in the “Recommending Potential miRNA” and “Proposing Efficient siRNA”. The overall workflow is shown in Fig. 1. In recommending efficient miRNAs and siRNAs, we aim to target UCRs. Thus, the UCRs of SARS-CoV-2 must be determined.

Figure 1 The overall workflow.

(1, 2) The unstructured conserved regions in the SARS-CoV-2 genome are found. (3) The miRNAs available in the CLASH database, mirBase, and bat-specific miRNAs are collected. (4) The most efficient miRNAs were selected based on their high binding affinity to the unstructured conserved regions of SARS-CoV-2. (5) Besides the selected miRNAs, several siRNAs are designed, which specifically target the desired regions in SARS-CoV-2. (6) The most potent rival targets for the selected miRNAs/siRNAs are analyzed. (7) Most efficient candidate miRNAs and siRNAs are introduced as potential therapeutics for COVID-19.

Extracting the UCRs in SARS-CoV-2

The conserved regions were obtained by Rangan, Zheludev & Das (2020). They conducted multiple sequence alignments (MSAs) on three sets of sequences to identify potentially conserved RNA regions. In the first MSA, they align the sequences gathered by Ceraolo & Giorgi (2020), including betacoronavirus whole-genome sequences. Then, 100% conserved continuous regions were extracted. The second MSA was done by executing the BLAST with highly similar sequences to SARS-CoV-2. The continuous regions with at least 98% conservation were identified. In the last MSA, a broad set of 180 complete betacoronavirus from the NCBI with 99% similarity to SARS-CoV-2 was considered. The continuous regions with at least 54% conservation in the MSA were extracted. The identified potentially conserved regions were filtered to consider only the regions with at least 15 nt. Furthermore, the potentially unstructured regions were determined by computing the unpaired probabilities. The base-pair probability matrix was calculated for the SARS-CoV-2 genome in the windows of length 120 nt (sliding by 40 nt) using Contrafold 2.0 (Do, Woods & Batzoglou, 2006). Then, the paired probability for every position is the average of base-pair probabilities in all windows that the position is included. Consequently, the positions with paired probability less than 0.4 were considered as the unpaired positions. UCRs of SARS-CoV-2 are denoted as the continuous stretches of unpaired positions with at least 13 nt. Fifty-eight unstructured conserved regions with lengths 13–26 nt were found. The entire list of UCRs in SARS-CoV-2 is provided in Supplemental Materials.

Recommending potential miRNA

Investigation for promising miRNAs is conducted according to the following steps:Computing the free binding energy for the hybridization of miRNAs of each dataset and UCRs of SARS-CoV-2: Binding energy is one of the important factors for measuring the affinity of binding (Khan et al., 2020). Effective binding between miRNA and its target is assessed using IntaRNA 2.0 (Mann, Wright & Backofen, 2017) in ViennaRNA library (Lorenz et al., 2011). The parameters were set to the default value. It considers both the interaction site accessibility and seed constraints in computing the free binding energy. The seed interactions must be genetically favorable, which is enforced in IntaRNA. Thus, analyzing minimum free energies (MFEs) computed by IntaRNA also incorporates site accessibility and seed match factors.

Analyzing the efficiency of utilizing the suggested miRNAs by investigating the biological functions related to the target UCRs of SARS-CoV-2.

Finding the most probable target mRNA in the human body that is a rival for target UCR and analyzing its GO terms and pathways.

Proposing efficient siRNA

Providing efficient and target-specific siRNA design for mammalian RNAi is done by siDirect 2.0 (Naito et al., 2009), which have the following steps to propose siRNA:We considered a window of length 23-mer and slide it on the target UCR in SARS-CoV-2. In this way, all possible 23-mer subsequences in the target sequence are considered, and the corresponding siRNA is designed with a 21 nt complementary guide and a passenger strand with two nt overhang at the 3’ end. Then, the list of all possible siRNAs was refined to the siRNAs that have the following three sequence conditions (Ui-Tei et al., 2004):

– with “A” or “U” nucleotide at the 5′ end of the guide strand

– with “G” or “C” nucleotide at the 5′ end of the passenger strand

– with more than 4 “A” or “U” residues in the 5′ terminal seven bp of the guide strand

– without contiguous substring of G/C of length larger than four bp

– with less than 30% GC content

Calculating melting temperature (Tm) for the seed-target duplex located at 2–8 from the 5′ end of the siRNA guide strand and its target (Ui-Tei et al., 2008). Tm calculation is done using the nearest neighbor model and the thermodynamic parameters for the formation of the RNA duplex (Naito et al., 2009).

Removing siRNAs that have near-perfect complementary to other human mRNAs, rather than the selected target location. siDirect analyzed 19 nt lengths from 2 to 20 positions in both strands via BLAST and pre-computed hits to find near-perfect match candidates (Yamada & Morishita, 2005). Finally, the selected siRNAs must have at least two mismatches with all non-targeted transcripts.

Results

Analysis #1: Suggesting candidate miRNAs from the human body for the treatment of COVID-19

To evaluate the efficiency of available miRNAs in the human body for targeting UCRs, we analyzed the CLASH database. CLASH is a gold standard dataset that contains experimentally validated interactions between human miRNAs and human mRNAs.

The goal of this analysis was to find appropriate available miRNAs that have efficient interactions with UCRs and low-affinity to human mRNAs. For each miRNA (M) in CLASH and each UCR (U) in SARS-CoV-2, the free binding energy of the hybridization of M and U is denoted by E(M,U) and computed by IntaRNA (Mann, Wright & Backofen, 2017). Moreover, the free binding energy of targeting each human mRNA (R) by miRNA (M) is denoted by E(M,R) is retrieved from CLASH and computed by IntaRNA. Among all mRNAs in human, the mRNA with the highest affinity to bind with miRNA is considered as the most potent rival in competition with UCR to be bound by miRNA. The efficiency score for targeting UCR by miRNA (denoted by S(M,U)) is calculated as (1).

(1) S(M,U)=E(M,U)−minR⁡{E(M,R)}

The negative efficiency score suggests that the affinity of miRNA to target that UCR is greater than targeting any mRNA in the body. Whereas the positive efficiency score denotes that there is at least one mRNA in the body that the miRNA has more propensity to target that mRNA, rather than targeting the UCR. Therefore, the lower the efficiency score is, the more favorable candidate the miRNA is. The analyzed miRNAs were sorted based on their efficiency scores, and the top 10 were considered as the candidate miRNAs. Using either the energies reported in CLASH or energies computed by IntaRNA result in the same list of candidate miRNAs.

Table 1 represents the most favorable candidate miRNAs. This table includes the information of candidate miRNAs, the information about target UCRs in SARS-CoV-2, and the most potent rival mRNAs. It can be seen that the candidate miRNA has more propensity to bind to UCR, rather than the rival mRNA. The entire report of this analysis, containing the interaction information for all miRNA in CLASH and all UCRs are presented in Supplemental Materials.

Table 1 The candidate miRNAs interactions.

The first column represents the mirBase ID of miRNAs. The three next columns contain information about target UCRs. The fifth column includes the UniProt ID of the most potent rival mRNA. The two next columns represent the free binding energy to target UCR and the most potent rival mRNA, respectively. The last column shows the efficiency score of the candidate miRNA.

miRNA	UCR sequence	UCR location	UCR region	rival mRNA ID	E(M,U)	minR{E(M,R)}	S(M,U)	
hsa-miR-374a-5p	UUACAAACAAUUUGAUACUU	19569–19588	ORF1ab, nsp11	ENSG00000186184	−15.27	−10.9	−4.37	
hsa-miR-548b-3p	GAAGAGCAACCAAUG	27361–27375	ORF6	ENSG00000109572	−10.05	−6.1	−3.95	
hsa-miR-1-3p	CACAUGCUUUUCCA	18681–18694	ORF1ab, nsp11	ENSG00000100485	−10.86	−8.4	−2.46	
hsa-miR-224-5p	UUUACUCAACCGCUACUUUAGAC	11659–11681	ORF1ab, nsp6	ENSG00000113273	−9.47	−7.1	−2.37	
hsa-miR-98-5p	UUUACUCAACCGCUACUUUAGAC	11659–11681	ORF1ab, nsp6	ENSG00000205250	−11.7	−10.7	−1	
hsa-miR-26a-2-3p	UCAAGAAAUUCAAC	28853–28866	ORF9, nucleocapsid protein	ENSG00000171772	−6.98	−6.3	−0.68	
hsa-miR-192-3p	UCUUGUCUGUUAAUC	16361–16375	ORF1ab, nsp13-ZBD	ENSG00000185658	−6.98	−6.7	−0.28	

The efficiency score of top 4 candidate miRNAs are less than −2. Thus, we conducted further analyses on these four miRNAs to investigate their targets in SARS-CoV-2. The hybridization of targeting UCRs by candidate miRNAs are shown in Fig. 2.

Figure 2 Hybridization of top candidate miRNAs with target UCRs.

Hybridization of top candidate miRNAs with target UCRs. (A) miRNA: hsa-miR-374a-5p,UCR location: 19569-19588, E(M,U)=−19.24kcal/mol. (B) miRNA: hsa-miR-548b-3p, UCR location: 27361–27375, E(M,U)=−16.03kcal/mol. (C) miRNA: hsa-miR-1-3p, UCR locatoion: 18681–18694, E(M,U)=−21.98kcal/mol. (D) miRNA: hsa-miR-224-5p, UCR locatoion: 11659–11681, E(M,U)=−22.23kcal/mol.

Among the potential efficient miRNAs, both hsa-miR-374a-5p and hsa-miR-1-3p have the most affinity to bind nsp11 in SARS-CoV-2. Protein nsp11 is an endoribonuclease that inhibits the production of IFN-β (Shi et al., 2011). The first essential factor of innate immune response to the viral infections is IFN-β (Weber, Kochs & Haller, 2004). Defeating the viral infection by host cells becomes more difficult due to the interferon antagonist property of nsp11 (Weber, Kochs & Haller, 2004; Bowie & Unterholzner, 2008). Hence, targeting nsp11 facilitates the immune response against SARS-CoV-2.

The second candidate miRNA is hsa-miR-548b-3p, which binds to ORF6 with low free binding energy. ORF6 is an accessory protein with key functions in viral pathogenesis (Yoshimoto, 2020; Kumar et al., 2007; Zhao et al., 2009). This protein has the most powerful suppression of interferon production and interferon signaling (Yuen et al., 2020). Moreover, it plays role in promoting RNA polymerase activity by interacting with nsp8 (Kumar et al., 2007). Consequently, silencing ORF6 is a promising approach in designing live-but-attenuated vaccines against SARS-CoV-2 (Yuen et al., 2020).

The fourth candidate miRNA, hsa-miR-224-5p, has the best binding with nsp6 region. Protein nsp6 produces autophagosomes, which assists in the assembly of replicase proteins (Yoshimoto, 2020). In addition, the function of nsp6 in limiting autophagosome/lysosome expansion, which leads to both inducing autophagy of host cells and preventing the autophagy of viral components (Lippi et al., 2020). By this way, nsp6 impedes the degradation of the viral components in lysosomes by prohibiting autophagosomes expansion (Yoshimoto, 2020; Cottam, Whelband & Wileman, 2014). As a result, down-regulating nsp6 improves host immunity by saving host cells from autophagy and degrading viral components.

Moreover, the most probable target genes for these miRNAs were obtained from CLASH (Helwak et al., 2013), and their related gene ontology (GO) and pathways were obtained from Uniprot (Apweiler et al., 2004). Figure 3 represents the GO biological process (BP), molecular function (MF), and cellular component (CC) of the target genes related to the top 4 miRNAs, as well as their corresponding pathways.

Figure 3 The GO terms and Reactome pathways related to the most probable targets of each candidate miRNAs from all species.

The first column shows the mirBase ID of the selected miRNAs. The three next columns show the GO terms for molecular function, biological process, and cellular component, respectively. The last column contains the related Reactome pathways.

Suggesting candidate miRNAs from the bats for the treatment of COVID-19

Bats are one of the most probable sources of SARS-CoV-2, and previous studies have shown that bats are more resistant to CoV infections (Hoffmann et al., 2013). Therefore, this idea comes up that maybe the bat-specific miRNAs have a functional role in their resistance against CoVs. The bat-specific miRNAs were investigated for efficient binding to UCRs to evaluate this hypothesis. The free binding energy for the hybridization of UCRs and bat-specific miRNAs were computed by IntaRNA (Mann, Wright & Backofen, 2017) tool. Since there is no gold standard for the interactions and free binding energy of bat-miRNAs and human mRNAs, the candidate miRNAs were selected based on the energy ratio. The energy ratio for each pair of miRNA (M) and UCR (U) is computed as formula (2).

(2) ER(M,U)=E(M,U)/E(M,M¯)

where E(M,M¯) is the free binding energy of the miRNA to its completely complement sequence. The bat-specific miRNAs were sorted based on their energy ratio, and the top 20 were selected as candidate miRNAs. The human mRNA targets for the candidate miRNAs were predicted using mirDB (Chen & Wang, 2020) tool. Table 2 represents the most favorable candidate miRNAs in bat, as well as their target UCR and their most probable predicted human mRNA by mirDB. The complete list of energy ratios for all bat-specific miRNAs and UCRS, as well as the entire list of predicted target genes for the selected bat-specific miRNAs are provided in Supplemental Materials.

Table 2 The candidate bat-specific miRNAs interactions.

The first column represents the sequence of bat miRNAs. The three next columns contain information about target UCRs. The two next columns represent the free binding energy to target UCR and the energy ratio, respectively. The last column includes the UniProt ID of the most potent rival mRNA.

Bat miRNA sequence	UCR sequence	UCR location	UCR region	E(M,U)	ER(M,U)	rival mRNA ID	
UGGCAAGUAGGUGAUAGGAUGU	UAUUCUGUUAUUUACUUGUAC	9578–9598	ORF1ab, nsp4	−21.37	0.5397	ENSG00000198964	
UGAGGUAGUAGAUUGUAUAGU	UUGUACUAAUUAUAUGCCUUAUUUCUU	6757–6783	ORF1ab, nsp3	−12.63	0.388615385	ENSG00000206557	
ACAAUUCUGUGUAUCUGAUC	UUAGAUAUAUGAAUUCA	11724–11740	ORF1ab, nsp6	−11.33	0.380328969	ENSG00000106153	
CGGGGGGUGGCGGGGAGGU	ACUCAAUUACCCCCUGCA	21626–21643	Surface glycoprotein	−19.23	0.411777302	ENSG00000141905	
AGAGGUAAAAAUUUGAUUUGACU	CACAAGUCAAACAAAUUUACAAA	23910–23932	Surface glycoprotein; spike protein, S2 glycoprotein	−11.42	0.37689769	ENSG00000133121	
GGGGCCGGGGGUGGGGGU	ACUCAAUUACCCCCUGCA	21626–21643	Surface glycoprotein; Spike protein	−16.46	0.374857663	ENSG00000127588	

Figure 4 displays the hybridization of targeting UCRs by candidate bat-specific miRNAs. The first three suggested bat-specific miRNAs target UCRs in nsp3, nsp4, and nsp6. These three proteins have transmembrane domains (Sakai et al., 2017). There is a fundamental relation between CoV infections and the nsp3 association (Khailany, Safdar & Ozaslan, 2020). Protein nsp3 is the most extended protein in SARS-CoV-2 with multiple domains, which plays crucial roles in forming the replication/transcription complex (RTC) (Lei, Kusov & Hilgenfeld, 2018). Due to essential protease activity for releasing proteins with viral activity, down-regulating nsp3 can be a desirable goal for antiviral activity (Báez-Santos, John & Mesecar, 2015). Protein nsp4 has an important interaction with nsp3, which induces membrane rearrangement and viral replication (Yoshimoto, 2020). This interaction is a critical factor in viral replication via the rearrangements of host-derived membranes. Elucidating this interaction leads to terminate SARS-CoV-2 replication (Sakai et al., 2017). Another transmembrane protein, nsp6, together with nsp3 and nsp4 proteins, configure the organelle-like replicative structures (double-membrane vesicles) (Cárdenas-Conejo et al., 2020). Making use of three first suggested miRNAs that target nsp3, nsp4, and nsp6 may strongly down-regulates the viral replication.

Figure 4 Hybridization of top 4 candidate bat-specific miRNAs with target UCRs.

(A) UCR location: 9578–9598, E(M,U)=−21.37kcal/mol. (B) UCR location: 6757–6783, E(M,U)=−12.63kcal/mol. (C) UCR location: 11724–11740, E(M,U)=−11.33kcal/mol. (D) UCR location: 21626–21643, E(M,U)=−19.23kcal/mol. (E) UCR location: 23910–23932, E(M,U)=−11.42kcal/mol. (F) UCR location: 21626–21643, E(M,U)=−16.46kcal/mol.

The next three suggested miRNAs have favorable affinities to bind the spike protein (surface glycoprotein). The entry of SARS-CoV-2 into cells is mediated by spike (S) glycoproteins, which binds to human angiotensin-converting enzyme 2 (ACE2) for cell entry (Hoffmann et al., 2020; Yoshimoto, 2020). The spike protein identifies the ACE2 protein on the surface of the host cell (Lan et al., 2020; Shang et al., 2020; Walls et al., 2020). The transmembrane spike (S) glycoprotein forms the homotrimers protruding from the viral surface (Tortorici & Veesler, 2019). This glycoprotein contains two subunits S1 and S2 that are essential for receptor binding and membranes fusion, respectively (Walls et al., 2016; Park et al., 2016; Kirchdoerfer et al., 2016). The S2 subunit is fusion machinery that facilitates the viral and cellular membranes fusion (Gui et al., 2017; Song et al., 2018; Yuan et al., 2017; Hoffmann et al., 2020). Since the surface glycoprotein S intervenes the virus entry into host cells, there is a major focus on this protein in therapeutic strategies and vaccine design (Walls et al., 2020; Hoffmann et al., 2020). Thus, the suggested miRNAs may have great antivirus activities via down-regulating the surface glycoproteins.

To analyze the possible side effect of using the candidate bat miRNAs in human body, we enriched their predicted targets in GO using UniProt (Apweiler et al., 2004). Figure 5 demonstrates the GO terms and Reactome pathways related to the most probable target mRNAs for the candidate bat-specific miRNAs.

Figure 5 GO terms and pathways for genes targeted by candidate bat miRNAs.

The first column shows the bat miRNA sequences. The three next columns show the GO terms for molecular function, biological process, and cellular component, respectively. The last column contains the related Reactome pathways.

Analysis #3: Suggesting potential miRNAs from 286 species for the treatment of COVID-19

The previous analyses suggested the favorable candidate miRNAs from human and bat that have a potential role in the treatment of COVID-19. Nevertheless, other species may contain miRNAs that reveal promising antiviral activities against SARS-CoV-2. To investigate this idea, we considered the miRNA sequences of all species available in mirBase (Kozomara, Birgaoanu & Griffiths-Jones, 2019). To choose the most effective miRNAs, we conducted the same analysis as the previous section, but on the miRNAs of numerous species. The selected miRNAs from all species are represented in Table 3. The entire list of analyzed miRNAs, as well as the list of the predicted targets for miRNAs are provided in Supplemental Materials.

Table 3 The candidate miRNAs from all species.

The first column shows the species name. The second column represents the sequence of miRNAs. The three next columns contain information about target UCRs. The two next columns represent the free binding energy to target UCR and the energy ratio, respectively. The last column includes the UniProt ID of the most potent rival mRNA.

Species scientific name	miRNA ID	UCR sequence	UCR location	UCR region	E(M,U)	ER(M,U)	rival mRNA ID	
Tribolium castaneum	tca-miR-6014-3p	GCUCUCACUCAACAUGG	28436–28452	ORF9ab, NAR region	−19.24	0.627323117	ENSG00000154447	
Monodelphis domestica	mdo-miR-7284b-5p	UUGUACUAAUUAUAUGCCUUAUUUCUU	6757–6783	ORF1ab, nsp3	−16.03	0.585891813	ENSG00000169855	
Ciona intestinalis	cin-miR-4020b-5p	UCUUUACCAACCACCACAAACCUCUAU	10009–10035	ORF1ab, nsp4	−21.98	0.584885577	ENSG00000213593	
Gallus gallus	gga-miR-1603	ACCAAACCAACCAUAUCCAA	6010–6029	ORF1ab, nsp3, NAR region	−22.23	0.581784873	ENSG00000185621	
Homo sapiens	hsa-miR-4500	AUAAGAAACCUGCUUCAA	6105–6122	ORF1ab, nsp3, NAR region	−13.99	0.570554649	ENSG00000206557	
Lotus japonicus	lja-miR7535	CCAUACCCACAAAUUUUAC	23700–23718	Surface glycoprotein; spike protein, Corona S2 glycoprotein	−19.98	0.566326531	ENSG00000134954	

The first candidate miRNA is for the red flour beetle (Tribolium castaneum) and targets a UCR in ORF9ab, which encodes nucleocapsid protein in SARS-CoV-2. It binds to the viral genome and provides stability for virus (Yoshimoto, 2020). It also has high expression during infection, which enables antibody responses (Peng et al., 2006). This protein is essential in viral RNA transcription and replication (Kang et al., 2020). Therefore, it is a promising target in UCRs for reducing viral transcription and replication.

Other candidate miRNAs are from gray short-tailed opossum (Monodelphis domestica), sea vase (Ciona intestinalis), red junglefowl (Gallus gallus), human (homo sapiens), and Lotus japonicus. These miRNAs target UCRs in nsp3, nsp4, and spike protein. As it is mentioned in “Suggesting Candidate miRNAs from the Human Body for the Treatment of COVID-19”, these regions are of high importance for viral transcription, replication, and binding to host cells. The suggested miRNAs from various species are potentially promising alternatives for targeting these important regions in SARS-CoV-2 and lowering its functionality and spread.

The hybridization of UCRs and selected miRNAs are displayed in Fig. 6. Furthermore, an investigation for possible side effects of using these miRNAs in human body was conducted by checking the GO terms and Reactome pathways of the rival mRNAs. The list of GO terms and Reactome pathways were obtained from Uniprot (Apweiler et al., 2004) and are exhibited in Fig. 7. Among the candidate miRNAs, tca-miR-6014-3 and hsa-miR-4500 are more favorable choices, since their molecular functions and pathways are related to defeating virus and improving immunity.

Figure 6 Hybridization of top candidate miRNAs with target UCRs.

(A) miRNA: tca-miR-6014-3p, UCR location: 28436–28452, E(M,U)=−19.24kcal/mol. (B) mdo-miR-7284b-5p, UCR location: 6757–6783, E(M,U)=−16.03kcal/mol. (C) miRNA: cin-miR-4020b-5p, UCR location: 10009–10035, E(M,U)=−21.98kcal/mol. (D) miRNA: gga-miR-1603, UCR location: 6010–6029, E(M,U)=−22.23kcal/mol. (E) miRNA: hsa-miR-4500, UCR location: 6105–6122, E(M,U)=−13.99kcal/mol. (F) miRNA: lja-miR7535, UCR location: 23700–23718, E(M,U)=−19.98kcal/mol.

Figure 7 The GO terms and Reactome pathways related to the most probable targets of each candidate miRNAs from all species.

The first column shows the mirBase ID of the selected miRNAs. The three next columns show the GO terms for molecular function, biological process, and cellular component, respectively. The last column contains the related Reactome pathways.

Analysis #4: Proposing efficient siRNA for the treatment of COVID-19

An efficient RNA-intervention technique is designing new efficient siRNAs. Since the length for siRNA is 21 nt, we considered all UCRs with minimum length of 21 nt. Then, the selected UCR sequences were given to the siDirect tool (Naito et al., 2009) for designing efficient siRNA to target these regions. Five siRNAs targeting four UCRs were found. The information of UCRs that are targetted by the designed siRNA is presented Table 4. The designed siRNA are listed in Table 5

Table 4 The selected UCRs for designing siRNA.

The length of these UCRs are at least 21 nt, for which significant siRNAs have been found.

UCR sequence	UCR location	UCR region	
UUGUACUAAUUAUAUGCCUUAUUUCUU	6757–6783	ORF1ab, nsp3	
CUGUCUUUAUUUCACCUUAUAAUU	17762–17785	ORF1ab, nsp13	
UCUUUACCAACCACCACAAACCUCUAU	10009–10035	ORF1ab, nsp4	
CACAAGUCAAACAAAUUUACAAA	23910–23932	spike protein	

Table 5 The designed siRNAs for the selected UCRs.

The first column indicates the location in SARS-CoV-2 that the designed siRNAs target these regions. The two subsequent columns contain the guide and passenger strands of the designed siRNAs. The next column shows the melting temperature for the seed-target duplex forming in the guide and passenger strands of the designed siRNAs. The next column denotes the satisfaction status of proposed siRNAs in the constraint of not containing more than 3 continuous “C”s or “G”s. The last column shows the “GC” content in the designed siRNAs.

UCR location	siRNA guide sequence	siRNA passenger sequence	Seed-duplex stability	Contiguous (G/C)	GC content (%)	
6760–6782	AAAUAAGGCAUAUAAUUAGUA	CUAAUUAUAUGCCUUAUUUCU	(10.9 °C, − 8 °C)	yes	<30	
6757–6779	UAAGGCAUAUAAUUAGUACAA	GUACUAAUUAUAUGCCUUAUU	(30.5 °C, 6.3 °C)	yes	<30	
17762–1774	UAUAAGGUGAAAUAAAGACAG	GUCUUUAUUUCACCUUAUAAU	(18.5 °C, 6.9 °C)	yes	<30	
10009–10031	AGAGGUUUGUGGUGGUUGGUA	CCAACCACCACAAACCUCUAU	(25.4 °C, 30.5 °C )	yes	<30	
23910–23932	UGUAAAUUUGUUUGACUUGUG	CAAGUCAAACAAAUUUACAAA	(−0.3 °C, 19.2 °C )	yes	<30	

The minimum free energy (MFE) foldings of these five siRNAs were obtained by RNAfold in the ViennaRNA package (Lorenz et al., 2011). The obtained foldings are depicted in Fig. 8. Moreover, the hybridization of the designed siRNA with the target UCR is gained using IntaRNa tool (Mann, Wright & Backofen, 2017) and shown in Fig. 9. In order to investigate the potential side effects, the most potent target mRNAs were calculated using the mirDB tool (Chen & Wang, 2020). The most likely targets for siRNA#1, siRNA#2, siRNA#3, siRNA#4, and siRNA#5 are ENSG00000213047, ENSG00000169241, ENSG00000070882, ENSG00000118263, and ENSG00000157106, respectively. The retrieved genes were inquired into Unitprot (Apweiler et al., 2004) for analyzing the Go terms and Reactome pathways. The GO terms and pathways related to the target genes of each siRNA are presented in Fig. 10.

Figure 8 The MFE folding of the designed siRNAs.

(A–E) Folding of potential siRNAs #1, #2, #3, #4, and #5, respectively.

Figure 9 The hybridization of designed siRNAs with target UCRs.

(A) The hybridization of the potential siRNA #1 and the UCR in location 6760–6782 with free binding energy = −25.5 kcal/mol. (B) The hybridization of the potential siRNA #2 and the UCR in location 6757–6779 with free binding energy = −28.18 kcal/mol. (C) The hybridization of the potential siRNA #3 and the UCR in location 17762–1774 with free binding energy = −29.57 kcal/mol. (D) The hybridization of the potential siRNA #4 and the UCR in location 10009–10031 with free binding energy = −38.97 kcal/mol. (D) The hybridization of the potential siRNA #5 and the UCR in location 23910–23932 with free binding energy = −24.4 kcal/mol.

Figure 10 The GO terms and Reactome pathways corresponding to the most probable targets of the deigned siRNAs.

The first column shows the proposed siRNA. Molecular function, biological process, and cellular component, are represented in the three next columns and Reactome pathway is shown in the last column.

Conclusion

The recent outbreak of severe acute respiratory syndrome coronavirus 2 (SARS-CoV-2) became pandemic and international concern. Currently, no vaccine or specific drug has been proposed for SARS-CoV-2 or other zoonotic coronaviruses. Therefore, it is essential and emergency to look deeper into potential treatments. RNA interference (RNAi) is a therapeutic strategy that is profitable in the treatment of viral infections when other conventional approaches fail to obtain promising results. This approach is based on delivering small RNA duplexes, including miRNA or siRNA, to the body for triggering the inhibition of specific genes. In this study, we investigated the potential RNAi-based therapy that may help in the treatment of COVID-19. We analyzed the efficiency of the available miRNAs as well as newly designed siRNAs for downregulating and silencing the regions in SARS-CoV-2 that are both conserved and unstructured with high probability. To this aim, the unstructured conserved regions in SARS-CoV-2 were obtained using multiple sequence alignment and computing the unpaired probability using a computational method. Fifty-eight unstructured conserved regions were obtained and considered as the desirable regions for targeting by miRNA/siRNA, due to their evolutionary conservation, disinclination to develop resistance, high affinity to bind by hybridization and more accessibility to therapeutic interventions (Rangan, Zheludev & Das, 2020).

Four independent analyses were conducted to investigate the application of RNAi-based therapies in the decline of SARS-CoV-2 survival. Three analyses aim to suggest potential miRNAs that exist in human, bat and 286 other species for downregulating the unstructured conserved regions of SARS-CoV-2. The investigation of efficient miRNAs was done using computing the free binding energy of the hybridization of miRNAs and the regions in SARS-CoV-2. The ones with the least free binding energy were considered as the candidate miRNAs. Among the investigated miRNAs, four human miRNAs, six bat-specific miRNAs, and six other miRNAs from other species were extracted that have a high affinity to bind essential and functional regions in SARS-CoV-2 genome. The candidate miRNAs were further examined to check their possible side effects. In order to do this, the most probable human mRNA targets for the candidate miRNAs were obtained from experimentally validated databases or computational tools. Moreover, the Go terms (containing molecular function, biological process, and cellular component), as well as Reactome pathways related to the most probable targets, were extracted. Among the recommended miRNAs, tca-miR-6014-3 is very promising because its side effects prevent viral activity in the body and reduce viral infections.

In addition to three mentioned analyses for suggesting potential miRNAs, another analysis was conducted to design efficient siRNAs for silencing the unstructured conserved regions of SARS-CoV-2. Five efficient siRNAs were identified for targeting four critical regions in SARS-CoV-2. These regions play essential roles in viral replication and the attachment of the virus to host cells. Therefore, the application of proposed siRNAs may inhibit the replication and entry of SARS-CoV-2. Moreover, the most potent human mRNA targets of these siRNAs were obtained using computational tools, and their GO terms and pathways were analyzed for investigating their possible side effects.

All in sum, this study recommends 16 potential miRNAs and five efficient siRNAs that may help in the treatment of COVID-19. The candidate miRNAs and siRNAs are promising cases to be validated experimentally in cultured cells, or mouse models can reduce the time and costs of trying non-promising cases.

Supplemental Information

Supplemental Information 1 The UCRs of SARS-CoV-2 and the results of Analysis 1-3.

Sheet 1: The UCRs of SARS-CoV-2 including their regions, sequences, corresponding proteins, and unpaired probabilities.

Sheet 2: The computed free binding energy between human miRNAs and mRNAs.

Sheet 3: Comparison of binding energy for binding human miRNAs to mRNAs or UCRs.

Sheet 4: The computed free binding energy between bat miRNAs and human mRNAs.

Sheet 5: The list of possible mRNA targets for candidate bat-specific miRNAs.

Sheet 6: The computed free binding energy between miRNAs of all species and human mRNAs.

Sheet 7: The list of possible mRNA targets for candidate miRNAs in other species.

Click here for additional data file.

The authors would like to thank Mahyar Fazelirad for his help in graphical designs.

Additional Information and Declarations

Competing Interests

Author Contributions

Data Availability

The authors declare that they have no competing interests.

Narjes Rohani conceived and designed the experiments, performed the experiments, analyzed the data, prepared figures and/or tables, authored or reviewed drafts of the paper, and approved the final draft.

Fatemeh Ahmadi Moughari conceived and designed the experiments, prepared figures and/or tables, authored or reviewed drafts of the paper, and approved the final draft.

Changiz Eslahchi conceived and designed the experiments, authored or reviewed drafts of the paper, and approved the final draft.

The following information was supplied regarding data availability:

Material and implementation are available at GitHub: https://github.com/nrohani/SARS-CoV-2.

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
