# Peer review of "DisCoVering potential candidates of RNAi-based therapy for COVID-19 using computational methods"

_PeerJ, doi:10.7717/peerj.10505_

## Round 0.1 · original submission · Minor Revisions

As you can see from the attached reviewer reports, there are small issues that the reviewers have with your manuscripts and that should addressed before the manuscript can be accepted for publication.

Reviewer 1 ·

Basic reporting

The manuscript is well motivated, introduced in a broadly understandable manner, and self-contained. Literature and background discussion is extensive and precise in detail.

Artwork and figures are very good if not perfect, except that tables like Fig. 3 could possibly be transmitted better, but that is a point the PeerJ technical office may work out together with the authors.

Data is shared and documented / linked.

Experimental design

Covid19 therapies is an area of immediate pressure, so of urgency and importance. siRNA/miRNA suppression therapies are among the promising approaches. Here the authors combine several established modern bioinformatics tool approaches are combined into a candidate-predictive workflow, as clearly summarized in Fig. 1.

The study is worked out and described in thorough detail.

There are no apparent ethical issues within this work visible.

As the authors narrow down to two suggested miRNA / siRNA any subsequent experimental studies would come up with a small number of experiments.

Validity of the findings

The study appears to be on high level and well worked out.

Predicting only 2 candidates might indicate that the acceptance thresholds in each step were quite tight. While I would wish the authors the success that 1 or 2 of them materialize into a working thereapy, I would encourage, if not, to extend the study to predict the next 20 candidates which might still be a manageable number to be tested.

Additional comments

As specified in the 3 evaluation areas (not forwarded to authors), this study appears to be well worked out and written conclusively.
The authors have set up an innovative prediction pipeline for candidates for RNAi based therapy.


As a footnote:
Predicting only 2 candidates might indicate that the acceptance thresholds in each step were quite tight. While I would wish the authors the success that 1 or 2 of them materialize into a working thereapy, I would encourage, if not, to extend the study to predict the next 20 candidates which might still be a manageable number to be tested.

·

Basic reporting

Line 32: CoV instead of COV.
Line 33&34: pls specify the date at which these numbers were taken.
Line 78 and 86: M Witkos et al., 2011).== pls check.
Line 94: You might mean Rous sarcoma?
Line 124: the human == delete the article "the".
Line 151: fref et al., 2016) == was that correct?
Line 162: Analyses instead of Analysis.
Line 163: These two procedure is === These two procedures are.
Line 164:overall workflow is shown in 1. pls clarify!!
Line 167: The conserved regions obtained by (Rangan et al., 2020). == Grammatical error. and here in this case, the Year is inside the parenthesis.
Line 214 &215: leave a space between the value and its identifies, I mean the Celsius degree.
Line 260: clash should be CLASH.
Line 262: you need a "," after GO.
Line 308: we conducted the analysis the same as the previous section: the sentence needs revision.
Line 311: mention the red flour beetle and put the scientific name in parenthesis.
Line 316: Also indicate the common names and put the scientific name in parenthesis.


--Did you mention Fig.1 and Fig. 2 anywhere in the text before their appearance?
--In the ligand of Fig. 2 4, and 6 the word location is mentioned three times>> did you mean location??

Experimental design

This article fall under the umbrella of Bioinformatics. it tries to establish an miRNA-based CoV targeting. the core question is reliable and well articulated. The analyses were done on scientific merit and the results showed good conclusion that could be utilized in future.

Validity of the findings

The findings of this article is of great importance as it approached molecular targeting of SARS-CoV-2 in reliable and efficient manner.

Additional comments

Dear Colleague,
Nice work and well-witten article.
however, some the results section needs more polishing especially for punctuations

---

## Round 0.2 · accepted · Accept

All previous reviewer comments have been addressed. As comments were only requiring minor revisions, I believe that your manuscript can be accepted for publication without another round of reviewing.